# Complication and Infection Risk Using Bone Substitute Materials to Treat Long Bone Defects in Geriatric Patients: An Observational Study

**DOI:** 10.3390/medicina59020365

**Published:** 2023-02-14

**Authors:** Jonas Pawelke, Vithusha Vinayahalingam, Thaqif El Khassawna, Christian Heiss, Larissa Eckl, Gero Knapp

**Affiliations:** 1Experimental Trauma Surgery, Faculty of Medicine, Justus Liebig University of Giessen, 35392 Giessen, Germany; 2Department of Trauma, Hand and Reconstructive Surgery, Faculty of Medicine, Justus Liebig University of Giessen, 35392 Giessen, Germany; 3Medical University Innsbruck, 6020 Innsbruck, Austria

**Keywords:** trauma, geriatric, osteoporosis, humerus, radius, tibia, bone substitute, low impact, falls

## Abstract

*Background and Objectives:* he treatment of large bone defects in geriatric patients often presents a major surgical challenge because of age-related bone loss. In such patients, the scarcity of healthy makes autologous grafting techniques hard to perform. On the one hand, clinicians’ fear of possible infections limits using bone substitute materials (BSM). On the other hand, BSM is limitless and spares patients another surgery to harvest autologous material. *Materials and Methods*: To address the aptness of BSM in geriatric patients, we performed a retrospective analysis of all patients over the age of 64 years who visited our clinic between the years 2011–2018. The study assessed postoperative complications clinically and healing results radiologically. The study included 83 patients with bone defects at the distal radius, proximal humerus, and proximal tibia. The defect zones were filled with BSM based on either nanocrystalline hydroxyapatite (NHA) or calcium phosphate (CP). For comparison, a reference group (empty defect, ED) without the void filling with a BSM was also included. *Results*: 106 patients sustained traumatic fractures of the distal radius (71.7%), proximal humerus (5.7%), and proximal tibia (22.6%). No difference was found between the BSM groups in infection occurrence (*p* = 1.0). Although not statistically significant, the BSM groups showed a lower rate of pseudarthrosis (*p* = 0.09) compared with the ED group. Relative risk (RR) of complications was 32.64% less in the BSM groups compared with the ED group. The additional beneficial outcome of BSM was demonstrated by calculating the number needed to treat (NNT). The calculation showed that with every six patients treated, at least one complication could be avoided. Radiological assessment of bone healing showed significant improvement in the bridging of the defect zone (*p* < 0.001) when BSM was used. *Conclusions*: In contrast to previous studies, the study showed that BSM could support bone healing and does not present an infection risk in geriatric patients. The NNT calculation indicates a wider potential benefit of BSM.

## 1. Introduction

Geriatric patients represent a common challenge in trauma surgery because of comorbidities and delayed healing processes. Recently, life expectancy has increased by an average of 3 months each year. While life expectancy at birth in 1964 was 68 years for males and 74 for females, it was 77.9 years for males and 82.5 for females in 2015 [1]. Many clinical trials reported an undisputable connection between aging and comorbidities [2]. For example, the elderly with Parkinson’s disease suffer more frequently from low-impact fractures [3]. Furthermore, fracture risk is associated with age-related signs such as low activity levels and poor nutrition and age-related diseases such as osteoporosis [4]. Therefore, comorbidities present a major risk for postoperative and in-patient complications that lead to prolonged hospital stays, which in turn increases the risk of inpatient mortality [5]. 

In surgical treatment, any foreign material can increase the risk of infection [6]. In trauma surgery, 1–2% are surgical site infections [7,8]. However, depending on the site of infection, a 55% risk for mortality is reported [9]. Therefore, to prevent possible infection resulting from BSM as foreign materials, it is discussed to use BSM mainly in young rather than geriatric patients.

However, infection healing complications in geriatric patients are more common than in younger patients. Barrey et al., 2021 reported increased levels of pseudarthrosis of dens fracture in geriatric patients. While patients younger than 70 years had 0% to 12.5% pseudarthrosis formation, others older than 90 years of age had 58.6% pseudarthrosis [10,11].

Although various clinical data for bone substitute materials in surgical treatment already exist, BSM assessment explicitly in geriatric patients remains scarcely addressed. Nonetheless, studies often define adulthood (older than 18 years) as a cutoff to their data without comparing young patients to older ones. Therefore, a search query in PubMed with the keyword “bone substitutes materials geriatrics” revealed no hits. A literature search using (Google scholar), revealed a single publication reporting the idea of using vault-filling materials to treat bone defects of stress fractures in geriatric patients [12]. Furthermore, the clinical trials website revealed three trials, two examining fixators (NCT03950349) and screws (NCT03807349) and one regarding filling tooth sockets with MBCP gel TM (NCT00740311).

This study investigated the use of bone substitute materials for treating bone defects in geriatric patients. In addition to the radiological course of bone healing, complications after surgery were evaluated to determine the safety of the bone graft substitute.

## 2. Materials and Methods

This retrospective observational trial enrolled patients who underwent surgical treatment of bone defects. Fractures of the distal radius, proximal humerus and proximal tibia were observed. To avoid the debatable definition of geriatric patients [13], we included patients with a minimum age of 65 years, as described in the British Geriatric Society. Only acute fractures were included in the study; pathological fractures were excluded (Figure 1).

Demographic data, age, sex, side of the fracture, and body mass index (BMI), were used to compare BSM-treated patients with ED and were obtained from patient files. Comorbidities were classified according to the American Association of Anesthesiologists score (ASA classification) [14]. Further, the newer Charlson-Comorbidity-Index (CCI classification) was implemented to differentiate existing comorbidities.

The severity of fractures was assessed in 2 categories. First, open fractures versus closed fractures., Open fractures are frequently associated with severe soft tissue damage and infection by contaminated wounds. Second, fractures were classified according to the Müller AO classification (AO classification) for fracture morphology.

Treatment, complications, and healing progression were radiologically examined (Figure 2). This is important to assess the performance and safety of the BSM as registered medical devices which are products with a medical purpose that are intended for use in humans. Although CT imaging can provide better and more detailed analysis, however, due to radiation laws in Germany, CTs are only performed when X-rays are not informative enough. Furthermore, the retrospective study limits the choice of methodology. In this study, each patient record was evaluated with up to 6 follow-up examinations (FU). To achieve a non-biased assessment, the radiographic double-blinded examination was done by 2 independent evaluators. To determine bone healing, a classification system was developed based on the previously reported criteria set by Bohnhof et al., Freyschmidt et al. and Islam et al. to achieve an enhanced evaluation of the fracture edges, fracture gaps, and articular surface [15,16,17]. Regarding the fracture margin, sharp edges (5), partially sharp and partially blurred edges (4), blurred edges (3), faintly visible edges (2), and no visible edges (1) were distinguished. The bridging process indicates the new bone tissue grows to fill the fracture gap and is assessed by its mineralization and localization in the gap. A visible edge was seen in fractures without consolidation (6). 

A reduction in density and a more impressive fracture gap (5) are followed by a blurred fracture line (4), a partially blurred and partially compressed fracture gap (3), and a compression of the fracture gap (2). The best pattern in the fracture gap is complete bridging with any fracture gap (1). Additionally, the implanted osteosynthesis material and local bone density were observed. The radiological assessment was analogous to the German school grading system.

The nature and number of complications were extracted from patient records. Infections are one of the potential complications that can occur when bone substitute materials are used to treat bone fractures. The risk of infection can increase due to a number of factors, including improper surgical technique, and contamination of the material or the surgical site. Treatment for an infection in the setting of a bone fracture with bone substitute materials may involve a number of different approaches, depending on the severity and cause of the infection. In mild cases, antibiotics may be sufficient to clear the infection and promote healing. In more severe cases, surgical intervention may be necessary to remove the infected tissue and any remaining bone substitute material. This may be followed by a course of antibiotics to prevent the recurrence of the infection.

In some cases, revision surgery may be necessary to remove the infected material and replace it with a new bone substitute material or autologous bone graft. This can be a complex and challenging procedure that requires a high level of surgical skill and experience.

Furthermore, delayed healing, ligament and muscle defects, CRPS and cartilage damage were evaluated. A delayed bone healing process was evaluated in 2 ways. According to the FDA definition of pseudarthrosis, non-union for more than 6 months without a healing process for more than 3 months defined pseudarthrosis. Furthermore, mal union was observed. Mal-union was defined as a continuous progression of bone healing criteria without completion within the first 6 months. The mortality rate could not be associated with the augmentation, and no deaths were reported until the end of the designated follow-up. In this study, we monitored delayed healing and secondary dislocation as indicators of fracture stability. In order to have a complete clinical evaluation, the presence of any neurological impairments was also assessed. These symptoms, which included ongoing intense pain, abnormal sensations such as tingling and numbness, and heightened sensitivity, persisted for over 6 weeks after the surgery.

Data analysis was performed using IBM^®^ SPSS^®^ Statistics (IBM Corporation, Armonk, NY, USA). The demographic data and complications were nominally scaled, and the radiological pattern was scaled ordinally. For comparing post-surgery outcomes, Mann–Whitney U test was used. The level of significance was set at ≤5%.

## 3. Results

The current retrospective study has the limitation of a non-random selection; since the patients in retrospective studies are selected based on specific criteria, the sample was not selected randomly, and sample size calculation was not appropriate and might result in bias. However, we have calculated the effect size based on the radiological data using Cohen’s d, which resulted in an effect size of 1.33 which is considered a medium-to-large effect size. This suggests that there is a statistically significant difference between the two groups in terms of bone healing, with the BSM group having a higher mean than the ED group.

We enrolled 106 patients older than 64 years [min:max; mean ± SD] [65:91; 75.31 ± 6.69]. (*n* = 45, 42.5%) patients underwent surgery without bone graft substitutes. Bone substitute material (BSM) was used to treat bone defects in (*n* = 61, 57.5%) patients with a quantity of [1.0:7.5; 2.1 ± 1.51] mL. (*n* = 42, 68.9%) were treated using a CP-based bone substitute (*n* = 19, 31.1%) by NHA augmentation. (*n* = 89, 84.0%) female and (*n* = 17, 16.0%) male patients were under evaluation. The left side was in (*n* = 63, 59.4%) cases affected, in (*n* = 43, 40.6%) the right side. The AO classification of fracture severity demonstrated (*n* = 14, 13.2%) extraarticular, Type A, (n = 15, 14.2%) partial intraarticular, Type B, and (*n* = 73, 68.9%) intraarticular, Type C, fractures. Most fractures were located at the distal radius (n = 73, 71.7%), with fewer at the proximal humerus (*n* = 6, 5.7%). The lower extremity, the proximal tibia, was affected in (*n* = 24, 22.6%). A loss of *n* = 4 (3.8%) preoperative radiographs was noticed. Classifying comorbidities, the ASA classification was 1.00:3.00 [2.28 ± 0.56], and the CCI was 2.00:13.00 [5.41 ± 2.26]. It should be noted that the CCI classification was evaluated in *n* = 74 patients (69.8%).

The BMI over all groups was [17.16:37.80; 26.84 ± 4.49] with a peak for patients with normal weight (BMI ≤ 24.9; *n* = 41, 38.7%) and mild adiposities (BMI ≤ 29.9; *n* = 39, 36.8%). Most injuries were caused by low-impact trauma (*n* = 87, 82.1%). High-impact injuries were noticed in *n* = 17 (16.0%), and at least *n* = 2 (1.9%) anamneses were missing. Subdividing into the type of injury, (*n* = 41, 41.5%) caused the injury by stumbling, followed by (*n* = 26, 24.5%) domestic falls. (*n* = 17, 16.0%) injuries were suffered by falls on stairs. High-impact injuries were subdivided into (*n* = 5, 4.7%) falls from a high point, (*n* = 2, 5.7%) sports injuries, and (*n* = 10, 9.4%) traffic accidents by car, bike, or motorcycle. Due to the clinical examination, (*n* = 13, 12.3%) patients were initially treated by external fixation. The typical duration of use of the external fixator until the change of procedure was [0:106; 14.92 ± 26.32] days. Antibiotic prophylaxis was administered in accordance with the hospital’s internal guidelines (*n* = 104, 98.1%). Mainly, a perioperative single-shot antibiotic, Cefuroxime, was used (*n* = 82, 77.4%). 

An important criterion in clinical evaluation after surgery is the survey of complications (Table 1). There was no relevant difference between the BSM and the ED group. Complications were noticed in the BSM (*n* = 21, 34.4%) and ED (*n* = 23, 51.1%) groups (*p* = 0.11). Regarding the number of complications, a wider range with a non-significant difference (*p* = 0.07) was seen in the BSM group [0.0:4.0; 0.43 ± 0.72] compared with the ED group [0.0:2.0; 0.64 ± 0.71]. Pseudarthrosis decreased in the BSM group (*n* = 3, 4.9%) compared with the ED group (n = 7, 15.6%) nonsignificant (*p* = 0.09) (Figure 3). Delayed healing, ligamentous and muscular damage, complex regional pain syndrome (CRPS), previous death before follow-up, secondary diseases and neurological diseases showed no significant difference (*p* > 0.05). In the evaluation for the risk of using bone substitute materials, infections were detected in the ED (*n* = 1, 2.2%) and BSM groups (*n* = 2, 3.3%). When comparing mal-union in the BSM and ED groups, a non-significant difference was found (*p* = 0.07). When summarizing the significance of complications, no difference was found. In the clinic, relative (RR) and absolute risk (AR) reductions of complications allow the use of new therapies. Bone graft augmentation resulted in an RR of 32.64%, an ARR of 16.68%, and an NNT of 5.99. 

Radiological assessment was used to determine bone healing and complications (Table 2). The margin of fracture showed significant differences in some FU examinations (FU3: *p* = 0.002; FU4: <0.001; FU5: <0.001). Furthermore, the fracture gap determined decreased numbers in the BSM group compared with the ED group (*p* < 0.05). The articular surface demonstrated an enhanced pattern in the BSM group (FU3–4: *p* < 0.05). Concerning the osteosynthesis material and the bone substance in the region of fracture, there was no difference between the groups (*p* > 0.05).

For comparing the BSM group with the CP, and the NHA group with the ED group, the data set was subdivided. Comparing the descriptive parameters, we found no difference between the ED and the CP group in age (*p* = 0.22), gender (*p* = 1.0), BMI (*p* = 0.33), and severity of fracture by the AO classification (*p* = 0.98). Furthermore, the region of fracture (*p* = 0.87) and the fracture of the collateral bone demonstrated no differences (*p* = 1.0). Comparing the descriptive parameters of the ED and the NHA group, there were no differences between the groups. Age (*p* = 0.28), gender (*p* = 0.28), BMI (*p* = 0.33), and the AO classification (*p* = 0.38) showed normal distribution between the groups. An increased number of fractures of collateral bones was located in the ED group (*n* = 31, 68.9%) compared to the NHA group (*n* = 6, 31.6%, *p* < 0.001). Furthermore, more open fractures were treated in the NHA group (*n* = 5, 26.3%) compared to the ED group (*n* = 1, 2.2%, *p* = 0.006). At least, the descriptive data between the CP and the ED group showed similarity. Age (*p* = 0.85), Gender (*p* = 0.99), BMI (*p* = 0.89) and the AO classification (*p* = 1.0) demonstrated well comparability. 

For daily clinical practice, the complications were compared between the three groups. At first, we compared the CP with the ED group. We saw fewer complications in the CP group (*n* = 12, 28.6%) than in the ED group (*n* = 23, 51.1%, *p* = 0.05). Non-union was seen less in the CP group (*n* = 1, 2.4%) compared to the ED group (*n* = 7, 15.6%, *p* = 0.06). Other complications, here infections (*p* = 1.0), mal-union (*p* = 1.0), CPRS (*p* = 1.0), cartilage damages (*p* = 0.58), and neurological diseases (*p* = 0.5), showed no differences between both groups. No statistical differences were seen between the NHA and the ED group. We noticed the same risk of suffering complications (*p* = 1.0) and similarity for non-union (*p* = 0.71), infections (*p* = 1.0), mal-union (*p* = 1.0), CRPS (*p* = 0.09), posttraumatic cartilage damage (*p* = 0.26), and neurological problems (*p* = 1.0). At least, there were no differences between the CP and the NHA group. We showed similarities in complications between both groups. Complications (*p* = 0.24), non-union (*p* = 0.23), infections (*p* = 1.0), mal-union (*p* = 1.0), CRPS (*p* = 0.1), posttraumatic cartilage damage (*p* = 1.0), cartilage damage (0.42), and neurological sequelae (*p* = 0.31).

Comparing the radiological criteria between the ED and the CP group, the fracture edge demonstrated decreased values in the CP group (FU1,2: *p* > 0.05; FU3: *p* = 0.002; FU4: *p* < 0.001; FU5: *p* = 0.002; FU6: *p* = 0.62). Evaluating the bridging process due to evaluating the fracture gap, significantly decreased grades were seen (FU1: *p* = 0.12; FU2: *p* = 0.03; FU3: *p* = 0.03; FU4: *p* < 0.001; FU5: *p* < 0.001; FU6: *p* = 0.15). Further on, grades of the articular surface decreased in the medial radiological follow-up examinations (FU1: *p* = 0.31; FU2: *p* = 0.08; FU3: *p* = 0.01; FU4: *p* = 0.01; FU5: *p* = 0.14; FU6: *p* = 0.11).

Biomechanically, there is a difference in bone healing between weight-bearing, the lower extremity, and the upper extremity. The subsequent data analysis of *n* = 82 (77.4%) fractures of the upper extremity was compared with *n* = 24 (22,6%) of the lower extremity. The descriptive parameters (mean upper extremity ± SD vs mean lower extremity ± SD, p-value), age (75.90 ± 7.10 vs 73.29 ± 4.60, *p* = 0.15), BMI (26.64 ± 4.27 vs 27.54 ± 5.25, *p* = 0.53), affected body side (right body side *n* = 34, 41.5%, left body side *n* = 47, 57.3% vs right side *n* = 8, 33.3%, left side *n* = 16, 66.7%; *p* = 0.46), added BSM (*n* = 49, 59.8% vs *n* = 12, 50.0%, *p* = 0.48), and ASA classification (ASA1 *n* = 6, 7.3%, ASA2 *n* = 45, 54.9%, ASA3, *n* = 31, 37.8% vs ASA1 *n* = 0, ASA2 *n* = 19, 79.2%, ASA3 *n* = 4, 16.7%, *p* = 0.22) demonstrated well comparability of the groups. Unequally distributed, most patients of the upper extremity were female (*n* = 73, 89%), with a significant difference to patients of the lower extremity with *n* = 16 (66.7%) female patients (*p* = 0.01). Furthermore, more fractures of the collateral bone and ulnar styloid were seen in the group of the upper extremity (*p* < 0.001).

The risk for complications was comparable between the groups (*p* = 0.48). Further on, there was no difference for individual complications between the group of the upper extremity and the lower extremity, non-union (*p* = 1.0), infection (*p* = 0.59), mal-union (*p* = 0.23), CRPS (*p* = 1.0), posttraumatic cartilage damages (*p* = 1.0), and neurological diseases (*p* = 0.59).

The radiological follow-up examinations demonstrated well comparability between the groups. The fracture edges (FU1: *p* = 0.21; FU2: *p* = 0.28; FU3: *p* = 0.74; FU4: *p* = 0.46; FU5: *p* = 1.0; FU6: *p* = 1.0) and the fracture gap (FU1: *p* = 0.49; FU2: *p* = 0.03; FU3: *p* = 0.92; FU4: *p* = 0.03; FU5: *p* = 0.82; FU6: *p* = 0.85) verified equality between the groups. The articular surface demonstrated significantly decreased numbers in the upper extremity (FU1: *p* < 0.001; FU2: *p* = 0.01; FU3: *p* = 0.03; FU4: *p* = 0.03; FU5: *p* = 0.003; FU6: *p* = 0.32).

## 4. Discussion

Complex fractures with substance defects in the bone represent an increasing challenge in trauma surgery [18]. Due to a steadily aging population, this situation will continue to worsen in the coming years. A study with data collection from 2009 to 2019 showed that 59% of the patients with fractures in Germany turned out to be 70 or older [19]. The goal of treating fractures in geriatric patients should be to minimize surgical stress while maintaining exercise or weight-bearing stability of the fracture. Significant progress has been made in this area recently with the use of stable-angle implants. In osteoporotic bone and larger defects, it is often impossible to achieve satisfactory stability intraoperatively. Additional defect augmentation with autologous bone graft is often out of the question in geriatric patients because of the additional risks with a donor-side morbidity rate of 15% [20]. As an alternative to allogeneic bone grafting, bone substitute materials have been insufficiently tested clinically to date. In our work, we could demonstrate the better performance of augmented fractures, particularly in the intermediate FU examinations (FU3 and FU4). As a hypothesis, already better stability could be interpreted here, which would represent an advantage in therapy, especially with regard to secondary dislocations or delayed bone healing.

Concerning complications, infection is well discussed in some case reports and trials [21]. Although the absolute number of infections was increased in the ABP group, there was equality in comparing the groups of treatment (*p* = 1.0). Further on, one of the most serious complications, pseudarthrosis, decreased (*p* = 0.09). If bone grafts are used in the case of pseudarthrosis, various trials demonstrate well healing processes [22]. Connecting these trials with the results of our study, bone graft substitute augmentation may prevent the risk of suffering a non-union.

Analyzing the subgroups of CP and NHA compared to the ED group, the groups demonstrated good demographic comparability. The in vivo safety of bone material was checked by comparing the complications between the groups. Fewer complications were seen in the CP group compared to the ED group (*p* = 0.05). Non-union, a severe complication in orthopedics, was decreased significantly (*p* = 0.06) compared to the ED group. There were no differences in complications between the ED and the NHA group and between the CP and the NHA group (*p* > 0.05). Summarizing this study, patients treated by CP augmentation benefit the most. Although there was no significant difference between the CP and the NHA group, patients with CP defect vault filling demonstrated fewer complications (CP: *n* = 12, 28.6%; NHA: *n* = 9, 47.4%). Previous histomorphology trials showed neither inflammation in using alloplastic bone substitute materials [23]. The clinical in vivo examination of the CP and the NHA groups provided this in vitro assumption in geriatric patients. Normally, geriatric patients suffer from chronic inflammation [24]. Instead of increased rates of infection caused by chronic inflammation in geriatrics, the same risk for infections was seen in the BSM groups (*p* > 0.05). Further on, both BSM groups showed advantages regarding the radiographic bridging process. Summarizing the data between the CP, the NHA and the ED group, geriatric patients benefit from using BSM. 

Prior trials have shown the various differences in biomechanics and bone healing between the upper extremity and the lower extremity [25]. Due to the body weight, the vascular neogenesis in fractures and bone vault filling could decrease. Suffering less vascular neogenesis, a delayed bridging process is possible. Comparing the upper and lower extremities in this trial, no differences in complications and in the radiological main outcome parameters were seen. The upper extremity demonstrated decreased grades in the articular surface. Concerning this, a well-bridging and healing process in using bone graft is demonstrated in the upper and lower extremity. NHA is a more refined form of CP and has been proven to give better results in terms of bone regeneration and biocompatibility [26]. However, despite the potential advantages of NHA over CP, through this study, there is no difference shown in terms of complication rates between the two BSM. Further clinical studies are needed to fully understand the differences between NHA and CP, particularly in terms of their impact on geriatric patients. 

The current study has some limitations that include the lack of calculated sample size, no comparable bone mineral density measurement for all patients, and the absence of CT scans. These limitations may affect the accuracy of the study results. Despite the aforementioned limitations, the study can still provide valuable information. Although it is a useful measurement, it is not the only factor that affects healing outcomes. Other factors that are addressed here, such as age, medical history, and treatment plan, play a role, especially in the descriptive context of the study. By examining factors that may have influenced healing outcomes, the retrospective nature of the study can still provide insights into the healing process and inform future treatment decisions.

Since our study is a retrospective data set, which was compared with a homogeneous comparison group, further investigations would have followed to verify this hypothesis. Regarding the questionable increased risk of infection when using bone substitute materials, we could not find any increased risk in this work. However, the statement about the risk of infection cannot be generalized. Due to the large quantity of bone substitute materials, this risk must probably continue to be assessed for the respective materials depending on their composition in clinical investigations.

## 5. Conclusions

Summarizing the presented data, bone graft substitutes are a safe alternative to autologous bone grafts. Multimorbid patients may benefit from bone substitute materials because of the lack of infection and a low NNT of 5.99 for preventing complications.

## Figures and Tables

**Figure 1 medicina-59-00365-f001:**
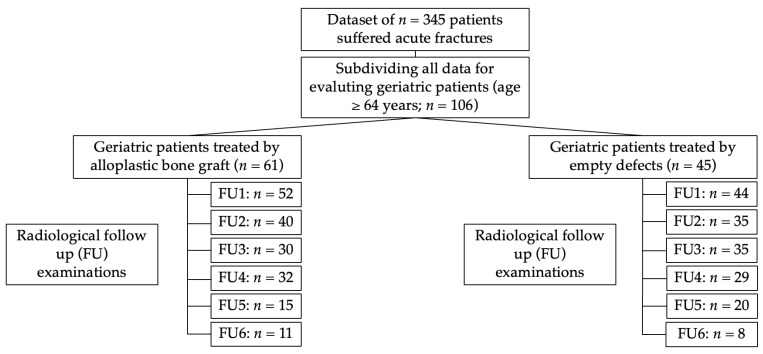
Overview of the patients treated with bone substitute materials compared with the empty cavity treatment.

**Figure 2 medicina-59-00365-f002:**
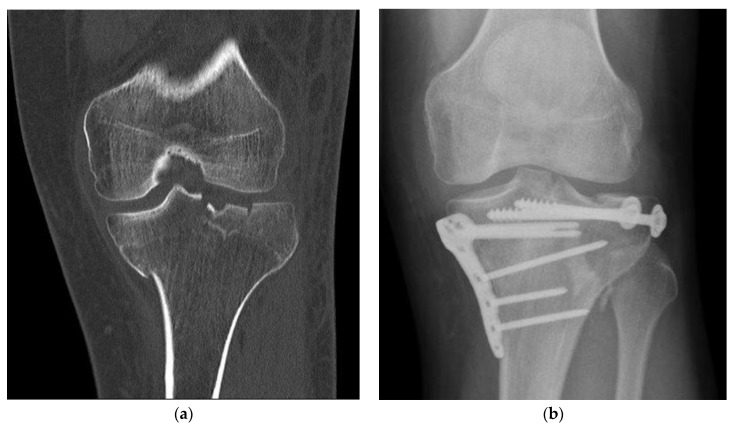
Fracture treatment of a proximal tibial fracture by surgical bone substitute material augmentation: (**a**) Preoperative CT scan demonstrates a metaphyseal compression fracture with intra- and extraarticular fracture gaps; (**b**) Alloplastic bone material substitute augmentation replaces the bone material of the fracture vault defects.

**Figure 3 medicina-59-00365-f003:**
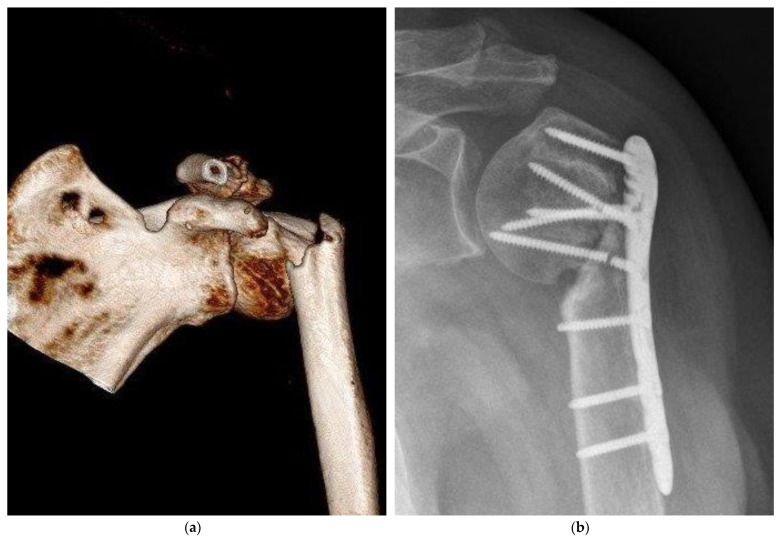
Severe complication, pseudarthrosis, resulting from a proximal humeral fracture, resulted by empty defect treatment. (**a**) Preoperative 3D CT reconstruction showed a sub-capital humeral fracture; (**b**) post-surgical care radiograph examination demonstrated an atrophic nonunion. Furthermore, two screws of the humeral head were broken.

**Table 1 medicina-59-00365-t001:** Overview of complications suffered in the groups using bone substitute grafting and empty defect treatment.

Complication	Bone Substitute Grafting	Empty Defect Treatment	*p*-Value (Mann-Whitney)
Complication (yes/no)	34.4%, *n* = 21	51.1%, *n* = 23	0.11
Number of complications	0 complications: 65.5%, *n* = 40 1 complication: 29.5%, *n* = 18 2 complications: 3.3%, *n* = 2 3 complications: 1.6%, *n* = 1	0 complications: 48.9%, *n* = 22 1 complication: 37.8%, *n* = 17 2 complications: 13.3%, *n* = 6	0.07
Pseudarthrosis (non-union)	4.9%, *n* = 3	15.6%, *n* = 7	0.09
Infection and necrosis	3.3%, *n* = 2	2.2%, *n* = 1	1.0
Delayed healing	0.0%, *n* = 0	2.2%, *n* = 1	0.43
Ligamentous and muscular defects	3.3%, *n* = 2	0.0%, *n* = 0	0.51
CRPS	3.3%, *n* = 2	0.0%, *n* = 0	0.51
Cartilage damage (arthrosis and chondromalacia)	11.5%, *n* = 7	20.0%, *n* = 9	0.28
Previous dead	1.6%, *n* = 1	0.0%, *n* = 0	1.0
Secondary dislocation	6.6%, *n* = 4	6.7%, *n* = 3	1.0
Neurological injuries	1.6%, *n* = 1	4.4%, *n* = 2	0.57
Mal-union	0.0%, *n* = 0	6.7%, *n* = 3	0.07

**Table 2 medicina-59-00365-t002:** Radiological assessment of fractures in traumatic caused geriatric fractures. Comparing the group of Empty Defect treatment (ED) with Bone substitutes material (BSM) in six follow-ups (FU) examinations.

Pattern of Fractures	Bone Substitute Grafting	Empty Defect Treatment	*p*-Value (Mann-Whitney)
Fracture edge	[3.00:5,00; 4.71 ± 0.54]	[3.00:5,00; 4.70 ± 0.55]	1
	[2.00:4,00; 3.30 ± 0.65]	[3.00:5,00; 3.49 ± 0.66]	0.424
	[1.00:5,00; 2.63 ± 0.79]	[2.00:5,00; 3.29 ± 0.86]	0.002
	[1.00:3,00; 1.60 ± 0.86]	[1.00:5,00; 2.59 ± 0.91]	<0.001
	[1.00:2,00; 1.07 ± 0.26]	[1.00:5,00; 2.10 ± 1.17]	<0.001
	[1.00:2,00; 1.18 ± 0.40]	[1.00:2,00; 1.38 ± 0.52]	0.603
Fracture gap	[2.00:6,00; 5.04 ± 1.04]	[3.00:6,00; 5.34 ± 0.96]	0.058
	[2.00:5,00; 3.33 ± 0.86]	[2.00:6,00; 3.92 ± 1.23]	0.020
	[1.00:5,00; 2.49 ± 0.88]	[2.00:6,00; 3.20 ± 1.30]	0.024
	[1.00:6,00; 2.00 ± 1.17]	[1.00:6,00; 2.62 ± 1.12]	0.004
	[1.00:2,00; 1.33 ± 0.49]	[1.00:6,00; 2.45 ± 1.40]	0.002
	[1.00:3,00; 1.36 ± 0.67]	[1.00:3,00; 2.00 ± 0.93]	0.12
Articular surface	[1.00:4,00; 2.85 ± 1.17]	[1.00:4,00; 2.89 ± 1.30]	0.547
	[1.00:4,00; 2.50 ± 1.36]	[1.00:4,00; 3.00 ± 1.26]	0.104
	[1.00:4,00; 1.97 ± 1.23]	[1.00:4,00; 2.77 ± 1.24]	0.013
	[1.00:4,00; 1.73 ± 1.08]	[1.00:4,00; 2.34 ± 1.11]	0.022
	[1.00:3,00; 1.60 ± 0.63]	[1.00:4,00; 2.10 ± 1.02]	0.163
	[1.00:2,00; 1.18 ± 0.40]	[1.00:3,00; 1.63 ± 0.74]	0.249

## Data Availability

Not applicable.

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
