# Peer review of "Complication and Infection Risk Using Bone Substitute Materials to Treat Long Bone Defects in Geriatric Patients: An Observational Study"

_medicina, 2023, doi:10.3390/medicina59020365_

Round 1
Reviewer 1 Report
Dear editor and author,
This is a well-prepared study that aims to investigate the use of bone substitute materials for treating geriatric fractures. It presents a clear method and adequate data processing for a definite outcome. Moreover, it is noted that the manuscript needs careful editing by someone with expertise in technical English editing. Here some significant flaws are as follows:
Comments 1:
Page 1-Abstract
“No differences in infections were found when comparing the groups (p=1.0). retrospectively evaluated.”
It is not clear in this sentence, please re-write this sentence.
Comments 2:
Page 1-Abstract:
The KEM and NNT should be written in the full name when they first appeared in the manuscript.
Comments 3:
Page 2
“Although various clinical data for bone substitute materials in surgical treatment already exist, substitutes in geriatric surgical assessment were never investigated. If one enters a search query in PubMed with the keyword "bone substitutes materials geriatrics" it remains unsuccessful. ”
I do not believe the topic is ever investigated, if the author could not find it in PubMed, do they check other databases?
Comments 4:
Page-3-M&M:
Does the author register this clinical trial in any official website, such as clinicaltrials.gov. ?
Comments 5:
Page-3-M&M:
Does the author calculate the sample size? What is the effect of this statistical analysis?
Comments 6:
Page-7-disscussion
“In 2006, half of the current traumas (30 to 40%) were caused by geriatrics [17]. ”
Please update the references in the discussion section.
Additionally, the discussion is too simple and inadequate, for example, there are no related paragraphs to discuss the outcomes in Table 2.
Author Response
The authors thank the reviewer for his work.
Page 1-Abstract
“No differences in infections were found when comparing the groups (p=1.0). retrospectively evaluated.”
It is not clear in this sentence, please re-write this sentence.
The authors thank the reviewer for noting the unclear statement and have changed the sentence accordingly
Comments 2:
Page 1-Abstract:
The KEM and NNT should be written in the full name when they first appeared in the manuscript.
The authors thank the reviewer for noticing the incorrect acronym KEM from the German first version. We have exchanged the uniquely used acronym and replaced it with the acronym NHA used in the paper, which was explained in the Introduction.
Comments 3:
Page 2
“Although various clinical data for bone substitute materials in surgical treatment already exist, substitutes in geriatric surgical assessment were never investigated. If one enters a search query in PubMed with the keyword "bone substitutes materials geriatrics" it remains unsuccessful. ”
I do not believe the topic is ever investigated, if the author could not find it in PubMed, do they check other databases?
The authors thank the reviewer for the idea of checking other data bases. We added one more data base without relevant information of bone substitutes in geriatrics.
Comments 4:
Page-3-M&M:
Does the author register this clinical trial in any official website, such as clinicaltrials.gov. ?
The work was designed retrospectively and an application was made to the relevant ethics committee prior to data collection. After a positive review, data collection was started. Registration in a database for clinical studies is only necessary for prospective studies.
Comments 5:
Page-3-M&M:
Does the author calculate the sample size? What is the effect of this statistical analysis?
The authors thank the reviewer for the annotation. The authors added the calculation of the effect size and added the limitation of the sample size.
Comments 6:
Page-7-disscussion
“In 2006, half of the current traumas (30 to 40%) were caused by geriatrics [17].
The authors thank the reviewer for pointing out a more current source and have corrected it accordingly.Between 2009 and 2019, the proportion of people over 70 years of age with fractures in the Federal Republic of Germany increased to 59%.
Please update the references in the discussion section.
Additionally, the discussion is too simple and inadequate, for example, there are no related paragraphs to discuss the outcomes in Table 2.
The authors thank the reviewer. We added more calculations in results and elongated the discussion based on more trials and the new calculation.

Reviewer 2 Report
- please explain in more detail about the complications in the form of infection and the treatment that might be done in addition
- What types of fractures are the research doing only in the joint area? Were any other types of fractures included in the study?
Author Response
The authors thank the reviewer for the comments.
- please explain in more detail about the complications in the form of infection and the treatment that might be done in addition.
The authors thank the reviewer for the statement. The authors added an explanation of infections and the treatment in material and methods.
- What types of fractures are the research doing only in the joint area? Were any other types of fractures included in the study?
The authors appreciate the statement. For explaining the severity of the fracture, the AO classification was used and insert in results. In order to make a comparison between the use of bone graft substitutes and osteosynthesis alone, the authors had to consider the fractures that were initially treated with bone graft substitutes in the control group. These were mainly non-significant higher classified, more complex fractures.

Reviewer 3 Report
Dear Authors,
This is an excellent article, BUT A FEW MAJOR CONCERNS HAVE TO BE ADDRESSED.
1. Materials and Methods 4th paragraph-first line- Complications and the radiological performance were evaluated to assess the safety of the medical devices. Please explain the term MEDICAL DEVICES.
2. Materials and Methods 4th paragraph-last but one line says- The fracture gap was controlled regarding the bridging process. Please rephrase the sentence to bring clarity to the readers.
3. Materials and Methods 6th paragraph's last part says- To control stability, delayed bone healing and secondary dislocation were under evaluation. Please rephrase/restructure the sentence to explain what the authors wanted to convey to the readers.
4. The authors have mentioned that the defect zones were filled with bone substitute material based on nanocrystalline hydroxyapatite (NHA) or calcium phosphate (CP). Clubbing both NHA and CP in the cohort is not advisable. Please compare NHA and CP groups separately with the void defects (ED) group.
5. The authors have selected the below-listed cohorts for the study.
distal radius (68.9%), proximal humerus (5.7%), and proximal tibia (22.6%).
Considering fractures in the non-weight-bearing upper limbs and weight-bearing lower limbs together for comparison with ED group may not give the true results as the bony architecture in the above groups varies significantly in the elderly population. Hence the cohort group needs to be reclassified into upper extremity and lower extremity groups and then compare each with the corresponding ED group.
6. Materials and Methods 4 th paragraph says fracture healing was assessed using XRAYS. A precise picture of bony healing may be obtained only by CT images. Why this modality was not used to check bony union in these elderly osteoporotic population?
7. Was BMD evaluation done prior to taking up patients into this study? A cohort with a variable BMD value cannot be included in a study that assesses the bony healing supplemented by BSM.
8. Along with question no.7, I feel it may not be correct to use such a cohort to compare the results between BSM and ED as the ED group will always show poor results if the low BMD falls into the ED group.
We look forward to hearing from you soon.
Thanks
Author Response
The authors thank the reviewer for all comments.
This is an excellent article, BUT A FEW MAJOR CONCERNS HAVE TO BE ADDRESSED.
- Materials and Methods 4th paragraph-first line- Complications and the radiological performance were evaluated to assess the safety of the medical devices. Please explain the term MEDICAL DEVICES.
The authors thank the reviewer and added the explaination of Medical devices which are products with a medical purpose that are intended by the manufacturer for use in humans.
- Materials and Methods 4th paragraph-last but one line says- The fracture gap was controlled regarding the bridging process. Please rephrase the sentence to bring clarity to the readers.
The authors rewrote the paragraph. In order to assess the progress of the consolidation, the fracture gap in particular was observed in the X-ray images. In the absence of bone healing, a visible edge was still present.
- Materials and Methods 6th paragraph's last part says- To control stability, delayed bone healing and secondary dislocation were under evaluation. Please rephrase/restructure the sentence to explain what the authors wanted to convey to the readers.
The authors thank the reviewer and explained the message of the sentence. As criteria for a stable fracture situation after surgical treatment, the parameters of delayed bone healing and secondary dislocation were examined.
- The authors have mentioned that the defect zones were filled with bone substitute material based on nanocrystalline hydroxyapatite (NHA) or calcium phosphate (CP). Clubbing both NHA and CP in the cohort is not advisable. Please compare NHA and CP groups separately with the void defects (ED) group.
The authors thank the reviewer for the excellent idea. The authors added a subgroup analysis.
- The authors have selected the below-listed cohorts for the study.
distal radius (68.9%), proximal humerus (5.7%), and proximal tibia (22.6%).
Considering fractures in the non-weight-bearing upper limbs and weight-bearing lower limbs together for comparison with ED group may not give the true results as the bony architecture in the above groups varies significantly in the elderly population. Hence the cohort group needs to be reclassified into upper extremity and lower extremity groups and then compare each with the corresponding ED group.
The authors added one more subgroup analysis to the section of results.
- Materials and Methods 4 the paragraph says fracture healing was assessed using XRAYS. A precise picture of bony healing may be obtained only by CT images. Why this modality was not used to check bony union in these elderly osteoporotic population?
The authors thank the reviewer for the statement. Unfortunately, as the study was retrospective, the authors were unable to influence this. Although CT scans were performed in some patients, no comparable data sets could be generated due to the irregularities in the data set.
- Was BMD evaluation done prior to taking up patients into this study? A cohort with a variable BMD value cannot be included in a study that assesses the bony healing supplemented by BSM.
As with note 7, we must also refer here to the limitations of the study due to the retrospective design. Examinations with regard to bone density by DEXA or QCT were not equally available in all patients. Therefore, a review regarding an equivalent distribution of BMD in both groups is not possible.
- Along with question no.7, I feel it may not be correct to use such a cohort to compare the results between BSM and ED as the ED group will always show poor results if the low BMD falls into the ED group.
Thanking the reviewer for the assessment, the authors added an subgroup analysis of the comparability between both BSM groups. Limiting, there is are smaller groups of the NHA and CP augmentation, the authors mainly check the safety of this medical device.

Round 2
Reviewer 1 Report
Dear editors and authors,
Thank you for the efforts on the revised manuscripts, I have no further questions after checking for the modified version.
Reviewer 3 Report
Thank you for your prompt and timely response .